Growth under cold conditions in a wide perennial ryegrass panel is under tight physiological control

Förster Lena 1
Grant Jim 2
Michel Thibauld 1
Ng Carl 3
Barth Susanne susanne.barth@teagasc.ie 1
1 Crops, Environment & Land Use Programme, Crops Research Centre Oak Park, Teagasc , Carlow , Ireland
2 Statistics and Applied Physics Department, Teagasc Research Operations Group , Dublin , Dublin , Ireland
3 UCD School of Biology and Environmental Science and UCD Earth Institute, University College Dublin , Dublin , Ireland
VanBuren Robert
Electronic publication date: 2018 Sep 11
Publication date: 2018
Volume: 6
Electronic Location ID: e5520
Received 2018 Feb 23; Accepted 2018 Aug 6
Copyright: ©2018 Förster et al.
Copyright year: 2018
Copyright holder: Förster et al.
License: This is an open access article distributed under the terms of the Creative Commons Attribution License, which permits unrestricted use, distribution, reproduction and adaptation in any medium and for any purpose provided that it is properly attributed. For attribution, the original author(s), title, publication source (PeerJ) and either DOI or URL of the article must be cited.
License URL: https://creativecommons.org/licenses/by/4.0/

Keywords: Cold stress, Perennial ryegrass, Acclimation to cold stress, Physiological control, Variation in germplasm

Funding: Teagasc Walsh Fellowship Postgraduate Scheme Research Stimulus Grant 14/S/81 Irish Department of Agriculture Food and the Marine (DAFM) Lena Foerster was supported by a studentship funded by the Teagasc Walsh Fellowship Postgraduate Scheme. This study is supported by a Research Stimulus Grant (Grant No: 14/S/81) funded the Irish Department of Agriculture, Food and the Marine (DAFM) to Carl Ng and Susanne Barth. The funders had no role in study design, data collection and analysis, decision to publish, or preparation of the manuscript.

==============================
Background

Perennial ryegrass is a cool-season grass species from the family Poaceae and is widely cultivated in temperate regions because it exhibits rapid growth and establishment, and possesses high forage quality. The extension of the growing season in Ireland in spring and autumn is a breeding target to make farming more profitable since a grass-fed diet based on grazing is the cheapest way of nutrition for ruminants.

Methods

Fifty-seven perennial ryegrass accessions were screened for their ability to grow under typical Irish spring conditions as taken from long term temperature records in controlled climate chambers. They were grown in low temperature (8 °C/2 °C day/night) and control conditions (15 °C/8 °C day/night) in three consecutive independent experiments. Fresh weight, height, chlorophyll content and electrolyte leakage were measured, and these parameters were used to rank plant performance under low temperature growth conditions.

Results

The results showed that height, yield and electrolyte leakage are excellent measures for the impact of cold stress tolerance. Little variation in growth was seen under cold stress, but a wide variety of responses were observed under control conditions.

Discussion

Our results suggest that cold stress is under tight physiological control. Interestingly, the various genotypes responded differentially to more amenable control conditions, indicating that a quick response to more amenable growth conditions is a better target for breeding programmes.

Introduction

Perennial ryegrass (Lolium perenne L.), a cool-season grass species from the family Poaceae with a self-incompatible and outcrossing nature (Cornish, Hayward & Lawrence, 1980). It is one of the most important perennial grasses worldwide (Wilkins, 1991). Perennial ryegrass is widely cultivated in temperate regions because it exhibits rapid growth and establishment, and it also possesses high forage quality (Casler & Duncan, 2003). In temperate grasslands an excess of forage production accumulates during the summer, but not sufficient fodder supply is available during the spring and autumn months emphasizing the economic value of early spring growth (McEvoy, O’Donovan & Shalloo, 2011). Ergon (2017) estimated that the prospects of improving spring growth are better than autumn growth. A breeding target for temperate grasslands is to improve spring grass growth. Temperature has a major impact on the growth of temperate forage grasses. Soil temperature was identified as the main determinant of growth in the South of Ireland. Developmental factors also determine the growth of grass, and productivity is highest in late spring and early summer, declining later in the summer (Hurtado-Uria et al., 2013). Often in spring a distinct peak of leaf extension is measurable. Seasonal effects on growth are demonstrated by transfer from cold into warm conditions in perennial and Italian ryegrass (Peacock, 1975; Parsons & Robson, 1980; Davies, Evans & Pollock, 1989). Low rates of growth were observed at temperatures down to 0 °C, whereas leaf elongation of perennial ryegrass increases strongly at temperatures above 5 °C (Peacock, 1975; Brereton, Carton & O’Keefe, 1985). However, to date there is little known about genotypic differences of perennial ryegrass in growth responses under cool conditions. Wilson (1975) demonstrated selection for genotypes within a variety of perennial ryegrass for low and high respiring genotypes at 8 °C and 25 °C, and Brereton & McGilloway (1999) studied varietal differences amongst eight varieties under natural cool temperate Irish spring and winter conditions. Cold or chilling stress results from temperatures cool enough (0 to 15 °C) to induce injury without forming ice crystals in plant tissues whereas freezing stress induces injury at temperatures below 0 °C. Plants from temperate climatic regions are considered to be chilling tolerant to variable degrees (Sanghera et al., 2011). Cold acclimatisation involves the remodelling of cells and tissues, and the reprogramming of metabolism and gene expression (Thomashow, 1999). Low temperature stress also inhibits various metabolic reactions resulting in altered phenotypic characteristics (Chinnusamy, Zhu & Zhu, 2007).

Limited solar radiation and short day length limit photosynthesis during the winter months in temperate grasslands which is resulting in source limitations. Once light conditions become more favourable, growth can be restricted by processes that inhibit cell division and expansion at low temperatures. Growth under these conditions becomes sink-limited (Wingler, 2015). Hormone signalling pathways play an important role in this regulation, in particular gibberellic acid (GA) signalling as GA also determines the growth of grass species, e.g., leaf extension (Stapleton & Jones, 1987), and in the induction of the fructan-degrading enzyme, fructan exohydrolase to promote growth after defoliation (Morvan et al., 1997).

The primary reactions of photosynthesis are temperature-independent and are catalysed by photosystem I (PSI) and photosystem II (PS II) to trap light energy and transform it into redox potential energy through a combination of photophysical and photochemical processes. This is leading to to charge separation. Temperature-dependent biochemical reactions convert this redox potential energy to stable reducing power (Ensminger, Busch & Huner, 2006). Low temperatures can inhibit electron transport by increasing membrane viscosity through alterations in the biophysical properties of thylakoid lipids and decreasing the rates of the enzymatic reactions involved in light absorption, energy transfer and transformation (Hüner, Oquist & Sarhan, 1998). A decline of chlorophyll content induced by low temperature can be seen as described e.g., by Rinalducci et al. (2011) who observed a reduction in of the chlorophyll content over time in spring wheat at low temperatures.

The objectives of this work were to investigate the extent to which natural variation towards cold tolerance under temperate temperature conditions in perennial ryegrass can be found, and how growth characteristics and stress indicators vary in different populations under cold and ambient control conditions.

Material and Methods

Plant materials

Fifty-seven forage accessions of perennial ryegrass (Lolium perenne) were used and they include commercial accessions, breeder’s seeds, and ecotypes from a range of geographical locations and breeding programmes (Table S1). The criteria used for selecting the accessions for experiments were adaptation to (1) temperate climates and (2) high land climates, and (3) a range of materials from world-wide breeding materials from different climatic zones.

Soil preparation

Soil used in all experiments was first sterilised and then mixed with 400 g of slow release nitrogen (38% N:38% ureic nitrogen), 400 g of nitrogen (27% N), 450 g of muriate of potash (50% K) and 400 g of superphosphate for 125 kg of soil. The soil mix (Westland Horticulture) consisted of 80% peat, 10% soil, 5% perlite, 5% grit, and was sterilised with steam for 48 hr before use.

Plant growth

Plants were sown in the glasshouse in a mini lawn, to simulate a meadow in a pot, at ambient conditions into fertilised soil and watered every two days and cut fortnightly. After 4 to 5 weeks, they were cut to 4 cm in height and transferred into climate controlled plant growth chambers (Microclima 1750; Snijders, Tilburg, The Netherlands), one pot of each accession into each chamber, namely one control chamber and one with cold settings. The experiment was repeated three times, Experiments 1, and 3 in 3 litre pots and Experiment 2 in 1 litre pots. Humidity settings and light intensity settings were 70% humidity and photosynthetic photon flux density (PPFD) of 320 µmol m−2 s−1, 12h/12 h light and dark in both chambers across all three experiments. The temperatures in the control chamber were 15 °C during the day and 8 °C at night. Temperature settings in the cold treatment chamber were 8 °C during the day and 2 °C at night. The cold temperatures were chosen to represent a very cool Irish spring day while the control conditions are typical of a typical Irish spring day (see Table S2). Plants were kept in the controlled environments for 73 days. The controlled environment chambers were monitored with a temperature and humidity sensor per chamber (Lascar EasyLog EL-USB-2) positioned between the plants. The experiments took place from November 2015 to August 2016 (Experiment 1: November 2015 to January 2016, Experiment 2: February 2016 to April 2016; Experiment 3: May 2016 to August 2016).

Parameters measured

Plant height was measured weekly over a period of 73 days. Height was determined from three points within the pot from the soil surface to the tip of the grass leaves. Chlorophyll content was measured on day 39 and day 63 using a SPAD chlorophyll meter (Spectrum Technologies Inc., San Jose, CA, USA) from five leaves within each pot by measurement of the optical density difference at two wavelengths, 650nm and 940nm. Electrolyte leakage was determined at days 40 and 70. Two leaves (5 cm long) were cut and then transferred to 25 ml centrifuge tubes containing distilled water and kept for 24 hr in the dark at room temperature. Conductivity of the solution was measured in µS before autoclaving at 121 °C for 20 min. The conductivity was measured again once the temperature was at ambient room temperature and taken as absolute conductivity to calculate electrolyte leakage using the equation shown below. electrolyteleakage=conductivityafter24hconductivityafterautaclaving∗100.

Fresh weights of the plants were determined at day 73 where all plants were cut to 4 cm in height before drying for three days at 70° to determine dry weight. Growth rate was determined by comparing the fresh weight of the plants under cold stress to those under control conditions.

Statistical analysis

Statistical analyses were conducted using SAS 9.4. A log transformation was applied to the raw data as most of the phenotypic data were not normally distributed (all experimental raw data are accessible as Supplemental Information). To determine if there were significant differences between cold and control treatments, a t-test at the 95% confidence level was used.

A general linear model analysis followed by a Tukey post-hoc test was applied in order to find groups with significant differences. The Glimmix procedure in SAS 9.4 was used, including interaction between treatment and experiment number in the model. The Tukey post-hoc test divides the accessions into groups where least squares means with the same letter are not significantly different for α = 0.05. Pearson correlations were calculated for the traits measured at the end of the experiments 1, 2 and 3.

Repeated measures analysis was used to examine changes over time for the height and the chlorophyll content values for all three experiments.

Growth curves were calculated with the Glimmix procedure by fitting the linear trend (y = a + bx) and also the quadratic model (y = a + bx + cx2) to test for evidence of curvature. Plots of raw data showed that the two treatments had a consistent pattern across days. Individual accessions appeared to show more variability in curve shape but in terms of the treatment groups this appeared to be negligible and simpler curves were fitted to describe the overall behaviour. A simple quadratic function was found to give a good fit to the data and clearly showed differences between treatment groups. The model fitted accounted for treatment, experiment and the response over time. The accessions do not appear explicitly in this model but they are included as subjects in a random coefficients analysis. The analysis fits an overall curve for the factors in the model and the subjects (each accession for each treatment for each experiment) are modelled as having the same form but randomly different from each other. All the subjects were combined to produce the overall curve and the deviations of the individual subject curve from the overall curve gives the measure of variance/error. Fitted coefficients were estimated and compared and residual checks were made.

Results and Discussion

The experimental conditions were chosen to represent typical late winter and early spring temperatures in Ireland. As seen in the climate data of Ireland (Table S2), the average daily maximum temperatures in January and February are around 8 °C, rising to nearly 15 °C in May. The average daily minimum for these months also corresponds with the chosen night settings. The cold stress was severe enough to effect differences in several of the measured plant traits between cold and control conditions (Table 1). Peacock (1975) demonstrated that temperature in the region of the stem apex exerted the greatest influence on the rate of extension of perennial ryegrass leaves. We thus deemed the chosen temperature regime as suitable to test the influence of cool temperatures on varietal spring growth. Additionally, the duration of day and night are similar in spring in Ireland, with the equinox on the 20th of March, hence plants were experimental grown under 12 hr day and 12 hr darkness.

Table 1 Median and interquartile range across the traits fresh weight (g), height (in cm), electrolyte leakage, and chlorophyll content.

Trait measured	Cold stress	Control	
	Median	Interquartile range	Median	Interquartile range	
Fresh weight	10.92	16.62	37.00	25.66	
Height (day 39)	11.00	4.75	21.00	13.67	
Height day (day)	13.67	4.54	28.00	8.75	
Electrolyte leakage day 40	9.52	5.52	8.18	3.66	
Electrolyte leakage day 70	11.58	3.35	7.68	4.20	
Chlorophyll content (day 39)	23.94	8.28	26.64	9.68	
Chlorophyll content (day 63)	31.98	11.32	32.82	5.92	

The fresh weight of the L. perenne accessions was lower under cold conditions (median 10.92 g) compared to plants grown under control conditions (median 37.00 g), and the interquartile range of the fresh weight was higher in the control than in the cold treatment (25.66 g compared to 16.62 g) (Table 1, Fig. 1). There was little variation amongst accessions under cold conditions and differences were not significant (Fig. 2). Wilson (1975) did not find in a diverging selection of two cultivar S23 groups any differences in winter growth. However, significant differences were observed between the different accessions under control conditions (p > 0.05), and the responses can be categorized into five groups according to a Tukey ranking test (Fig. S1). Wilson (1975) reported on a good broad sense heritability for growth in low and high respiring S23 cultivar genotype groups under amenable 25 °C day time temperatures. Substantial differences in the fresh weight between the experiments were found. This suggests a notable sensitivity of perennial ryegrass to small differences in temperature.

Figure 1 Box and whisker plot for plant height at day 67 (A) and fresh weight at day 73 (B), and electrolyte leakage day 40 (C) and day 70 (D) under cold and control treatment.

The results are derived from three experiments.

Figure 2 Fresh weight of the 57 accessions of perennial ryegrass under cold treatment and control conditions at day 73.

The results are from three separate experiments (experiment 1, 2 and 3).

Plant heights measured on day 67 were significantly different between the accessions grown under cold (13.67 cm) and control (28.00 cm) conditions (Tables 1 and 2). The plants grown under control conditions also exhibited a higher interquartile range of 8.75 cm compared to 4.54 cm under cold conditions (Table 1 and Fig. 1). There were two significantly different groups for plant height under cold stress and three significantly different groups under control conditions (Fig. S2). In relation to leaf extension under cold and control conditions, it would be interesting to know how the 57 accessions differed in their allelic polymorphism of the gene for the DELLA protein GAI involved in the gibberellic acid pathway which has been demonstrated before to explain differences in leaf elongation (Auzanneau et al., 2011).

Table 2 Results of Analysis of Variance (Anova) for accession, treatment and accession by treatment interactions.

Trait	Pr > F	
	Accession	Treatment	Accession by treatment	
Fresh weight in g	<0.0001	<0.0001	0.9986	
Height day 67 in cm	<0.0001	<0.0001	0.6862	
Electrolyte leakage day 70	0.1663	<0.0001	0.4568	
Chlorophyll content day 63	0.0460	0.5279	0.4668	

Electrolyte leakage from leaves at day 40 (9.52) and day 70 (11.58) were higher for plants grown under cold conditions compared to plants grown under control conditions at day 40 (8.18) and day 70 (7.68) (Table 1, Fig. 2). These effects were statistically significant for the accessions and the treatments, however not for accession by treatment interactions (Table 2). There were two Tukey test ranking groups for electrolyte leakage which overlap for the two treatments (Fig. S3). Higher electrolyte leakage value at day 70 compared to day 40 in the cold stressed leaves suggests an increase of cell membrane damage or loss of membrane integrity under cold stress conditions. In general, higher electrolyte leakage in the cold stressed leaves provides an indication of membrane damage occurring even at temperatures above 0 °C. However the increase in electrolyte leakage values could also be a consequence of higher electrolyte accumulation at lower temperatures known as cold hardening (Gay & Eagles, 1991).

The chlorophyll content at day 63 exhibited a greater variability for plants grown under cold conditions with an interquartile range of 11.32, compared to 5.92 for plants grown under control conditions (Table 1). However, differences were not significantly different under cold and control conditions (Table 2; Fig. S4). For 23 out of 57 accessions, chlorophyll content was higher in the cold stress conditions. In aging leaves, sugar accumulation can down-regulate photosynthetic gene expression and accelerate leaf senescence (reduction in green colour), but this response is abolished in cold acclimation resulting in a longer photosynthetic lifespan (summarized by Wingler, 2015). This effect might have been expressed in the 23 more green accessions in this study. Photosynthesis could be further investigated in relation to stomatal conductance and chlorophyll conductance measurements. Combining these measurements enables the collection of information on photosynthetic performance of the accessions under cold stress conditions.

No correlations were found for chlorophyll content, height, and fresh weight under control conditions. Additionally, there were no correlations between chlorophyll content and electrolyte leakage (Table 3). We observed significant correlations between plant height and fresh weight under control conditions. Negative correlations were observed between electrolyte leakage and chlorophyll content under control conditions. We observed positive correlations between fresh weight, plant height, chlorophyll content, and electrolyte leakage under cold conditions. Plant height under cold conditions was positively correlated with chlorophyll content and electrolyte leakage (Table 3). There was a positive correlation between fresh weight and height with electrolyte leakage although higher electrolyte leakage values which could be due to cold hardening (Gay & Eagles, 1991). Also lipidomics approaches could be used to assess changes in composition of cell membranes under cold since similar effects like in winter rye in cold hardening could be observed (Lynch & Steponkus, 1987). The fresh weight is considered as the most important trait in this study because yield is in general the most important trait for farmers. The height is highly correlated with the fresh weight with a correlation coefficient of 0.85 under control conditions, and a similar correlation coefficient of 0.88 under cold stress conditions. Additionally, height measurement data from earlier time points during the growth period can be used as a good proxy for the fresh weight at the end of the growth period.

Table 3 Pearson correlation coefficients for measured traits, fresh weight, height (at day 67), chlorophyll content (at day 70), and electrolyte leakage (at day 70) (significance is indicated below the coefficients).

Correlations for cold treatment are in bold; correlations for control treatment are not in bold.

	Fresh weight	Height day	Chlorophyll content	Electrolyte leakage	
Fresh weight		0.85	0.14	0.09	
		***		***	
Height	0.88		0.14	0.22	
	***			**	
Chlorophyll content	0.58	0.54		−0.47	
	***	***		***	
Electrolyte leakage	0.24	0.25	0.03		
	***	***			
Notes.

*** = 0.001; ** = 0.01; * = 0.05.

The analysis fits an overall curve for the factors in the model and subjects (each accession for each treatment for each experiment) are seen as having the same form in experiments 2 and 3, but are randomly different from each other. Experiment 1 differed from experiments 2 and 3 in shape (Fig. 3). Individual accessions appeared to show more variability in curve form but in terms of the treatment groups this variation appeared to be negligible and simpler curves were fitted to describe the overall behaviour (Fig. 5). A simple quadratic function was found to give a good fit to the data and clearly showed differences between treatment groups. The model fitted accounted for treatment, experiment and the response over time. The analysis fits an overall curve for the factors in the model and accounts for subjects (each accession for each treatment for each experiment). Experiment as a factor was found to interact with treatment and with the fitted coefficients for the time response and there were also interactions between treatment and the fitted coefficients, resulting in significantly different curve shapes across the experiments (Fig. 3). The coefficients, linear and quadratic, for each treatment are all significant which means that there is a linear trend and curvature for both treatments. When comparing coefficients for treatments a stronger trend (greater magnitude and more positive) for the control and stronger curvature (greater magnitude and more negative) for the control were found.

Figure 3 Response over time comparisons for plant height (cm) at days 0, 35 and 65.

(The accessions are included as subjects in the analysis); A: exp 1 = experiment 1, B: exp 2 = experiment 2, C: exp 3 = experiment 3.

The growth differences between cold treatment and an ambient control at 15 degree C in experiments 2 and 3 were comparable to differences in growth observed at similar temperatures by Peacock (1975). The growth response in experiment 1 in the control treatment was much lower and had a different shape. This could be due to being carried out in a different season compared to experiments 2 and 3 which might have impacted on the run performance of the control chamber. Hurtado-Uria et al. (2013) reported that besides temperature having an effect of growth also other factors like evapotranspiration having major effects on growth. Plants being sensitive organisms would recognize slight differences in season (Inoue, Araki & Endo, 2018). All of these factors would impact on plant growth models under varying temperature regimes. The experiments were run sufficiently long (73 days) to see long term effects of cold temperatures. To mimic field conditions, the grass were sown in mini lawns to simulate competition of the single plants for nutrients, light and water as they would experience in the field. Additionally, the mini lawns were grown in the glasshouse for up to five weeks, and cut twice prior to the start of the treatments to simulate a grazing regime and to ensure of well-established plant materials going into the experiments.

Growth under cold conditions did not show much variation in a wide panel of 57 perennial ryegrass accessions. The picture was different under more amenable control conditions. It is likely as outlined by Peacock (1975) and Davies, Evans & Pollock (1989) that developmental factors influenced by short pulses of enhanced temperatures and sun conditions lead to enhanced leaf extension in spring, a hypothesis supported by a pilot study on winter growth in eight accessions of perennial ryegrass which found improved growth in early heading varieties over the winter (Brereton & McGilloway, 1999).

Conclusions

We conclude that growth under cold conditions as in our experiment is under tight physiological control which did not allow in the 57 accessions from a wide variety of origins any variation in growth response under cold conditions as experienced in Ireland in a typical spring season. However looking at more amenable temperatures as experienced during a warm spring period, large differences in growth amongst the 57 accessions can be found. Future directions of research should be to repeat the cold stress experiments by supplementing the experiment by a cold treatment with short pulses of enhanced temperatures and sun/light conditions to identify germplasm which can make most use out of additional external growth factors. Desirable would be also to learn about the transcriptional control of excellent responding and poor responding germplasm to cold conditions. It would be also worthwhile to investigate the action of a combination of gibberellic acid and brassinosteroids as it has been shown in Arabidopsis may contribute to growth under cold by stimulating cell division and expansion (Fridman & Savaldi-Goldstein, 2013; Kim et al., 2010; Unterholzner et al., 2015) in grasses. It has been shown that brassinosteroid signalling is important in determining growth in grass species (Thole et al., 2012). Some of these interactions could be elucidated from a well-planned transcription expression experiment.

Supplemental Information

Table S1 The fifty-seven accessions of perennial ryegrass, their origin, ploidy status, maturity group and further information

Click here for additional data file.

Table S2 Irish weather data averages from Met Éireann (The Irish Meterological Service, http://www.met.ie)

Irish weather data averages from Met Éireann (The Irish Meterological Service, www.met.ie ) for January, February, March, April and May of the weather stations and years: Malin Head 1981–2010, Kilkenny 1978–2007, Shannon Airport 1981–2010, Dublin Airport 1981–2010, Birr 1979–2008, Belmullet 1981–2010, Clones 1978–2007, Cork Airport 1981–2010, Mullingar 1979–2008, Rosslare 1978-2007 and Valentia 1981–2010.

Click here for additional data file.

Figure S1 Fresh weight (g) at day 73 for the 57 accessions grown under cold stress and control conditions

The letters represent significant different groups according to an ANOVA test and a Tukey ranking test. Both accession and treatment were highly significant.

Click here for additional data file.

Figure S2 Plant height (cm) at day 67 for the 57 accessions grown under cold stress and control conditions

The letters represent significant different groups according to an ANOVA test and a Tukey ranking test. Both accession and treatment were highly significant.

Click here for additional data file.

Figure S3 Electrolyte leakage at day 70 for the 57 accessions grown under cold stress and control conditions

The letters represent significant different groups according to an ANOVA test and a Tukey ranking test. The factor treatment was highly significant.

Click here for additional data file.

Figure S4 Chlorophyll content at day 67 of the 57 accessions grown under cold stress and control conditions

The letters represent significant different groups according to an ANOVA test and a Tukey ranking test. Neither factor, accession or treatment, was significant.

Click here for additional data file.

Figure S5 (A) Plotted growth height raw data of experiment 3 for control and cold treatment and (B) modelled growth height data of experiment 3 for control and cold treatment

Click here for additional data file.

Data S1 Raw data for all traits measured in experiments 1, 2 and 3 for 57 perennial ryegrass accessions

Click here for additional data file.

The authors acknowledge the contributions of intern students to physiological measurements: Annabelle Gilgen, Tarah Kearney, Dominik Petrovic, Shirley Bernard and Emma Power. We are indebted to Teagasc and Goldcrop (Ireland), Agroscope (Switzerland), IBERS (UK) and Barenbrug (The Netherlands) for supplying us with seeds. We acknowledge careful recommendations of Prof. Marcel Jansen (UCC) and Dr. Wieland Fricke (UCD) in improving the drafting of this manuscript.

Additional Information and Declarations

Competing Interests

Author Contributions

Data Availability

The authors declare there are no competing interests.

Lena Förster performed the experiments, analyzed the data, contributed reagents/materials/analysis tools, prepared figures and/or tables, authored or reviewed drafts of the paper, approved the final draft.

Jim Grant analyzed the data, contributed reagents/materials/analysis tools, prepared figures and/or tables, authored or reviewed drafts of the paper, approved the final draft.

Thibauld Michel performed the experiments, contributed reagents/materials/analysis tools, prepared figures and/or tables, authored or reviewed drafts of the paper, approved the final draft.

Carl Ng conceived and designed the experiments, contributed reagents/materials/analysis tools, authored or reviewed drafts of the paper, approved the final draft.

Susanne Barth conceived and designed the experiments, performed the experiments, contributed reagents/materials/analysis tools, authored or reviewed drafts of the paper, approved the final draft.

The following information was supplied regarding data availability:

The raw data are provided in a Supplemental File.

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
