# Peer review of "Growth under cold conditions in a wide perennial ryegrass panel is under tight physiological control"

_PeerJ, doi:10.7717/peerj.5520_

## Round 0.1 · original submission · Major Revisions

Your manuscript has been seen by three qualified reviewers. Based on their detailed assessments and my own, I feel this manuscript could be suitable for publication in PeerJ after a number of major revisions. In addition to addressing each of the reviewer comments, the authors should improve the clarity of their manuscript in their revision.

Reviewer 1 ·

Basic reporting

Literature review on hormone signalling and gene expression during cold stress is fragmented and not relevant to the study conducted.

Experimental design

Description off the Material and methods is not sufficient. No indication how often chlorophyll content was measured. Authors state that dry weight of the plants was determined however no data provided and discussed.

Validity of the findings

The conclusion that growth under cold conditions is under tight genetic control is not validated by authors' findings. On contrary authors state that they could not see any variation in growth response amongst 57 accessions under cold conditions. Furthermore authors indicate that cell membrane damage occures at temperatures above 0 degrees Celcius by providing higher electrolyte leakage at day 70 compared to day 40. Increase in electrolyte leakage could also be a consequence of higher electrolyte accumulation at lower temperatures known as cold hardening.

Additional comments

The work presented provides a solid initial step for further in-depth analysis of genetic control of perennial ryegrass growth under low temperature conditions. However the manuscript presented in it's present form has major experimental flaws which should be addressed first.

Reviewer 2 ·

Basic reporting

The English Language is OK, but some sentences should be improved.

Literature, background, context:
The objectives stated at the end of the Introduction is: 1) to find out how much natural variation in cold tolerance (more precisely: chilling tolerance) there is in perennial ryegrass and 2) how growth characteristics vary between ambient vs chilling temperatures and in different genotypes. (comment: it is not genotypes, but populations, that are compared). The growth characteristics studied are shoot biomass accumulation (above 4 cm), leaf elongation, chlorophyll content and electrolyte leakage (i.e. not only growth characteristics).

The background is that with climate change it is a goal to breed for varieties that better utilize the extended growing season. In this respect it is relevant to refer to Ergon, Agronomy Journal, 2017 (Optimal Regulation of the Balance between Productivity and Overwintering of Perennial Grasses in a Warmer Climate). Is there no probability of freezing damage or chilling damage if Growth starts to early in spring under Irish conditions? Maybe not, but I miss a judgement of that. There is interesting and relevant information in the Introduction, but it seems a bit disconnected from the objective of the study, the experiment and the results. The introduction describes some processes relevant for cold acclimation, but this is not considered much further in the manuscript, which focusses on Growth and stress symptoms at chilling temperatures. Did you do a literature search on "(perennial ryegrass) AND (low temperature)"? I suspect you would find more literature that is more specifically related to Your study and objectives. I also think there is more relevant literature on Growth models for perennial ryegrass.

The results are not sufficiently discussed in relation to other scientific literature. Only very few References are mentioned in the Results&Discussion, and there they are mostly presented as statements, without being linked properly to the discussion of the presented results.

Structure
Results and Discussion is combined into one Chapter, unlike the recommendation from PeerJ.

Results vs hypotheses/objective
The results and objectives fit together - The (lack of ) variation in Growth at low positive temperatures is shown, and the effect of low positive termperatures on growth and stress symptoms is characterized. This is what the paper should focus more on, both in the title, introduction, and discussion.

Experimental design

No comment, seems OK

Validity of the findings

Results from Statistical analysis is not reported in a satisfactory way. I would like to see tables With degrees of freedom, significance Levels etc. Would it be better to do an ANOVA With Population, treatment and the interaction between these?

I am Critical to the title, and the conclusion that growth under Cold conditions are under "tight Genetic Control". First of all, I am not sure what this is supposed to mean, but to me it indicates that Genetic Control is more important than environmental Control. I think the results shows the opposite: there is environmental Control, but no Genetic variation. The proposed idea of selecting for improved Growth under warm pulses is interesting and relevant, although the Reference to Brereton and McGilloway appears a bit far-fetched, and had better been placed in the discussion before conclusions. The rest of the cocnluding paragraph should also be moved out of the conclusions, and the conclusions should rather focus on answering to the objective of the study.

Additional comments

State if you are using forage cultivars, lawn cultivars, or both.

I think the results are interesting, but the paper is too immature, and not sufficiently focussed yet.

Reviewer 3 ·

Basic reporting

The mansucript is interesting but not well written and the message poorly packaged.

The authors should be encouraged to improve the English - many sensences are complex with the sub-clauses often in the wrong place for ease of comprehension.

The major problem, in my opinion, is that the results are presented as if they dont mean much. The authors *may* have found evidence for a differential brake on growth but their dataset is not sufficient to allow them to make this claim. For this they would need to collect data for at least some strains at intermediate temperatures; alternatively, they should review their existing data in the light of current ambient temperature literature and gibberellin literature, and critically revise the hypotheses that could be then generated. They could also suggest how they could improve or extend their current expts tto test those hypotheses.

there is significant mechanistic understanding (and gaps) in terms both of ambient temperature response and in growth arrest due to stress from model systems such as Arabidopsis and rice. Given the general conservation of mechanism across the tree of life, it would seem very likely that these pathways would be very highly conserved in flowering plants.

Experimental design

A key problem is that they have tested only 2 temperatures. its probably not feasible, nor fair , to ask for more in this report. However, there is no serious discussion of how the study could be extended to address this - and also explain clearly and concisely why it is necessary. Such discussion should be added.

Validity of the findings

they seem fine - in so far as they go.

Additional comments

The response you report in interesting but by not relating it well to the current thinking on growth arrest under stress, you under-sell the story.

Conversely, the title of the paper seems to over-claim. Based on the current data, i dont think you can make the statement "under tight genetic control". While i'd like to think you are correct, and you might be, you've only tsted a small number of environments and you dont provide an indication of what the genetic mechanisms are or even might be.

---

## Round 0.2 · accepted · Accept

Based on advice from the reviewers and my own detailed assessment, I feel this revised manuscript is now suitable for publication in PeerJ.

# Reviewer 1 ·

Basic reporting

No comment

Experimental design

No comment

Validity of the findings

No comment

Additional comments

Authors have taken efforts to address the areas of concern and have substantially improved the manuscript.